# Genome-Wide Identification and Expression Profiles of C-Repeat Binding Factor Transcription Factors in *Betula platyphylla* under Abiotic Stress

**DOI:** 10.3390/ijms241310573

**Published:** 2023-06-24

**Authors:** Xiang Zhang, Jiajie Yu, Ruiqi Wang, Wenxuan Liu, Song Chen, Yiran Wang, Yue Yu, Guanzheng Qu, Su Chen

**Affiliations:** State Key Laboratory of Tree Genetics and Breeding, Northeast Forestry University, Harbin 150040, China; nefuzhangxiang@163.com (X.Z.); 13514554189@163.com (J.Y.); yjswrq@outlook.com (R.W.); liuwenxuan19990114@163.com (W.L.); chensongnet@gmail.com (S.C.); wangyiran810027843@163.com (Y.W.); yu666888yue@163.com (Y.Y.)

**Keywords:** *Betula platyphylla*, CBF transcription factor family, bioinformatic analysis, ABA, salt treatment, cold treatment, transcription activating activity

## Abstract

CBF (C-repeat binding factor) transcription factor subfamily belongs to AP2/ERF (Apetala 2/ethylene-responsive factor) transcription factor family, known for playing a vital role in plant abiotic stress response. Although some CBF transcription factors have been identified in several species, such as *Arabidopsis*, tobacco, tomato and poplar, research of CBF focus mainly on model plant *Arabidopsis* and have not been reported in *Betula platyphylla* yet. In this study, a total of 20 BpCBF subfamily members were identified. The conserved domains, physicochemical properties, exon-intron gene structure and the structure of conserved protein motifs of BpCBFs were analyzed via bioinformatic tools. The collinearity analysis of CBF genes was performed between *Betula platyphylla* and *Arabidopsis thaliana*, *Betula platyphylla,* and *Populus trichocarpa*. The *cis*-acting elements in the promoter region of *BpCBF*s were identified, which were mainly environmental stress-related and hormone-related element components. In this case, the expression patterns of the 20 *BpCBF*s upon ABA or salt treatment were investigated. Most of these transcription factors were responsive to ABA or salt stress in different plant tissues. The up-regulation trend upon cold treatment of the six cold-responsive genes validated by qRT-PCR was consistent with the result of RNA-seq. BpCBF7 showed transcription activating activity. This study sheds light on the responses of BpCBFs to abiotic stress and provides a reference for further study of CBF transcription factors in woody plants.

## 1. Introduction

Plant growth is affected by various abiotic stresses, such as salt, drought, high or low temperature and heavy metal stress. Facing the unpredictable or continuous adverse conditions, they have gradually developed multiple strategies for survival during long-term evolution, one of which refers to transcriptional regulation. Transcription factor, as a kind of protein, plays a key and central role in the process of transcriptional regulation by specifically binding to the *cis*-acting elements in the promoters of genes, and thereby activates or represses the transcription of these genes [1]. AP2/ERF (Apetala 2/ethylene-responsive factor) transcription factor family is one of the largest plant-specific transcription factor families and is marked by an AP2 conserved domain [2]. This transcription factor family is divided into five subfamilies according to the number of AP2 domains contained and sequence traits, including ERF (Ethylene Responsive Factor), AP2, DREB (Dehydration-Responsive Element Binding), RAV (Abscisic Acid Insensitive 3/Viviparous 1) and soloist [3]. CBF transcription factor subfamily belongs to the DREB subfamily. CBF1, a member of CBF (C-repeat binding factor) transcription factors, was first separated in *Arabidopsis* from cDNA library via a yeast one hybrid experiment by Stockinger et al. [4,5,6]. CBF transcription factors can specifically bind to the CRT/DRE sequence in the promoters of downstream genes and thereby activate or inhibit the expression of these genes, most of which are related to stress response, especially cold stress [7].

CBF subfamily has been mostly studied in *Arabidopsis*, in which six members have been identified. *AtCBF1*, *AtCBF2* and *AtCBF3* are tandemly located on Chromosome 4 [8]. They shared high similarity in gene sequences and some overlapping functions in plant cold stress responses [9]. Overexpression of each of the three genes, respectively, can promote the tolerance of *Arabidopsis* against cold, drought or salt stress, while the repression of *AtCBF1* or *AtCBF3* reduced the tolerance of *Arabidopsis* against cold stress [9,10]. *AtCBF4* was reported to be responsive to drought stress but not to cold stress. Overexpression of *AtCBF4* in *Arabidopsis* raised the expression levels of DRE-containing downstream genes, which function in plant cold or drought stress responses [11]. The expression of another two members, *AtDDF1* and *AtDDF2* were induced by drought and salt stress [12,13]. Ectopic expression of *Arabidopsis CBF* genes in other plants [14,15,16,17,18,19,20,21,22] or other plants’ *CBF* genes in *Arabidopsis* [23,24,25] both improved the cold tolerance of transgenic plants, indicating that the function of CBF in different species may have shared characteristics. Additionally, CBF transcription factors also play important roles in other abiotic stress responses, such as salt and drought stress. In banana (*Musa acuminata*), CBF transcription factor *MaDREB1F* was responsive to salt and osmotic treatment and expressed at the highest level at 24 d, 17 d, respectively [26]. Overexpression of *MaDREB1F* in banana enhanced drought tolerance of the transgenic plants. In *Kandelia obovate*, *KoCBF3* was induced by salt treatment (reaching a maximum of 10-fold at 3 h) [27]. This gene showed significant down-regulation after PEG treatment (reaching a minimum of 0.04-fold at 48 h). In apple (*Malus baccata*), *MbCBF1* was induced by salt (reaching a maximum of over 20-fold at 12 h) and drought (reaching a maximum of over 10-fold at 5 h) treatment [28]. Overexpression of this gene in *Arabidopsis* improved high salt tolerance of the transgenic plants.

To date, CBF subfamily members have been identified and characterized in many other species. Ding et al. identified six CBF subfamily members in *Populus trichocarpa* by analyzing the transcriptomic data with *Arabidopsis* sequences as reference [29]. Lee et al. identified ten CBF subfamily members in *Brassica rapa* by searching the genome database [30]. The increased quantity of subfamily members in *Brassica rapa* compared to that in *Arabidopsis* can be attributed to genome triplication and reorganization. Furthermore, eight genes of the ten were significantly up-regulated after cold treatment, and some of them were also responsive to salt, drought and ABA treatment. In *Taraxacum kok-saghyz*, ten CBF subfamily members were identified and their basic properties and relationships were analyzed with bioinformatic methods [31]. It was also found that the expression levels of most of the CBF subfamily members in *Taraxacum kok-saghyz* were significantly changed after cold treatment. Wang et al. identified 12 CBF subfamily members in *Lolium perenne* and divided them into three phylogenetic clades [32]. The expression levels of these genes were significantly up- or down-regulated in different patterns after cold, heat, water deficit, salinity and ABA treatment, and were more influenced by cold treatment. Otherwise, CBF transcription factors were also identified, and their functions were characterized in many other species, such as rice, barley, wheat, tobacco, tomato, grape and eucalypt [33,34,35,36,37], but are rarely reported in woody plants.

Birch, as a widely spread tree species across the globe, is well known not only for its ornamental value, but also its remarkable adaptability and tolerance to stresses. It possibly has developed mature strategies to survive the adverse environmental conditions. In this study, a total of 20 CBF transcription factors were identified in *Betula platyphylla* via the analysis of genomic data. With bioinformatic tools, we analyzed their protein sequences, chromosome locations, evolutionary relationships, basic physiochemical properties and gene structures. The expression patterns of these genes in different plant tissues upon salt, ABA and cold treatment were also investigated using qRT-PCR. This study hopes to provide a reference for future research on CBF transcription factors and plant stress response.

## 2. Results

### 2.1. Identification and Sequence Analysis of CBF Transcription Factors in Betula platyphylla

The *Betula platyphylla* v1.1 genome database was searched by BlastP with the six AtCBFs as query sequences. After Pfam analysis, HMMER prediction and another two conserved domain screening, a total of 20 BpCBFs were identified (Table 1). They were named BpCBF1-BpCBF20 according to their location order on chromosomes. The physicochemical properties of the 20 putative BpCBF family transcription factors were analyzed, and the length of their protein sequences ranged from 155 to 332 amino acids (aa). The physicochemical properties, such as chromosome location, isoelectric point and molecular weight, were listed in Table 1.

The phylogenetic tree of the CBFs in *Betula platyphylla*, *Arabidopsis thaliana* and *Populus trichocarpa* was constructed (Figure 1). The six AtCBFs were divided into two clades, which was consistent with the results of previous studies [30]. In order to more intuitively clarify the evolutionary relationships between BpCBFs, we introduced the members of CBF transcription factor family in the woody model plant *Populus trichocarpa* into the phylogenetic tree as well. The six PtCBFs were divided into three clades, which was consistent with the results of previous studies [29]. BpCBF14-20 were divided into Clade 2, indicating that the family members were highly conserved in the evolution process. BpCBF5 and BpCBF13 were closely related with PtCBF3 and PtCBF4, which were clustered in Clade 3. Others member, BpCBF1 -BpCBF4 and BpCBF6 -BpCBF12, were classified in Clade 1.

The phylogenetic tree of BpCBFs, which was constructed using the TBtools-II (v1.120) software based on the full-length protein sequences of BpCBFs, was constructed to further investigate the conservatism and evolutionary relationships (Figure 2A), which was consistent with the result in Figure 1. The gene structure (exon-intron) analysis (Figure 2B) of the twenty BpCBF transcription factor family members showed that only three CBFs (BpCBF9, BpCBF10 and BpCBF20) contained only one intron, with an intron length longer than 2 kb. None of the other members contain introns. Except for BpCBF20 which contained 5′-regions, other members contain neither 5′-regions nor 3′-regions. Based on the results of the online MEME tool motif analysis, schematic representations of the structures of all BpCBF proteins were constructed (Figure 2C,D). A total of 10 conserved motifs were obtained. Motif 1 and Motif 3 were distributed in all transcription factors, and Motif 3–6 exists in almost all members. Transcription factors in the same class usually had similar motif structures. For example, BpCBF14 -BpCBF20 had roughly the same motif structures (Motif 8), while BpCBF1 -BpCBF13 did not contain this motif, which might be an evolutionary difference. The 10 motif sequences were shown in Figure 2E. Their width, E-value and conserved domain were labeled in Appendix A.

### 2.2. Multiple Sequence Alignment Analysis of the 20 BpCBFs

It was reported in previous studies that CBF protein has three conserved domains, including AP2 domain, Special Domain I or Special Domain II [8]. In this study, a multiple sequence alignment of the 20 BpCBFs was performed, and the complete analysis results were shown in Appendix A. All of these conserved domains were present in the 20 BpCBFs (Figure 3). The three domains were continuously aligned, with Special Domain Ⅰ and Special Domain Ⅱ on the left and right side of AP2 domain, respectively. They were located at around positions 275- 353aa of the sequences.

### 2.3. Chromosome Distribution and Analysis of BpCBF Transcription Factors

The chromosome location of the 20 BpCBFs were mapped based on the genomic information of *Betula platyphylla* (Figure 4). The results showed that these 20 genes were located on three different chromosomes (Chr02, 10, 11). BpCBF1-12 were tandem duplications on Chr02, BpCBF14-20 were tandem duplications on Chr11 and BpCBF13 alone located on Chr10. That these family members were located in different chromosome positions was consistent with the phylogenetic cluster in Figure 1, which indicated that the members on the same chromosome were a tandem result caused by gene duplication in the evolution process, and also showed the stability of protein conservative structure in the evolution process.

### 2.4. Collinearity Analysis of BpCBF Transcription Factors

In order to explore the duplication mechanism of BpCBF transcription factor family, we constructed comparative syntenic maps between birch and *Arabidopsis* and birch and poplar, respectively (Figure 5). The results showed that the *BpCBF* had more syntenic transcription factor pairs with the *PtCBF*, than with *AtCBF*. The number of syntenic CBF transcription factor pairs between birch and Arabidopsis was two (Appendix A) and between birch and poplar was five (Appendix A). *BpCBF14* had collinear genes in the other two plants, and there were collinear sites in the other two species, and only *BpCBF14* was collinear with the *AtCBF*s. The collinear *CBF*s between *Betula platyphylla* and *Arabidopsis thaliana* and *Betula platyphylla* and *Populus trichocarpa* are displayed in Appendix A.

### 2.5. Analysis of Cis-Acting Elements in the Promoter Regions of BpCBFs

We identified 21 *cis*-acting elements that play key roles in plant growth by analyzing the 2000 bp sequences upstream the BpCBFs CDS. Analysis showed that these *cis*-acting elements included light-responsive elements, site-binding-related elements, development-related elements (Figure 6). And most environmental stress-related and hormone-related element components can be found in the promoter regions of BpCBFs, most frequent ones including light-responsive element (280), abscisic acid response element (ABRE, 199), MeJA-responsive element (99), Myb-bingding site (80) and stress response element (STRE, 60). Other elements like gibberellin response element (GARE), low-temperature response element, and other components can also be found (Appendix A). Taking ABRE as an example, the promoter of BpCBF10 had the most ABRE, the number of which is 15. The promoter of BpCBF9 contained 14 ABRE, 14 for that of BpCBF11, −13 for that of BpCBF5. It showed that BpCBFs may be closely related to plant abiotic stress response.

### 2.6. Tissue-Specific Expression Analysis of BpCBFs

To elucidate the expression patterns of BpCBFs, the relative expression levels of these genes in roots, stems and leaves of birch were measured using qRT-PCR (Figure 7). Overall, most of these genes were expressed at the highest levels in leaves, lower in stems and lowest in roots. Additionally, the difference degree between the relative expression levels of each gene in different tissues diverges between these genes. For instance, the relative expression level of BpCBF8 in leaves was 2.25 times that in stems and 7.55 times of that in roots. However, BpCBF20 was nearly expressed at the same level in stems and leaves, which is approximately three times that in roots. Apart from this trend, different expression patterns did exist. For instance, BpCBF6 and BpCBF19 had the highest expression levels in stems, lower expression levels in leaves and lowest expression levels in roots. BpCBF17 had the highest expression levels in roots, lower expression levels in leaves and lowest expression levels in leaves. The total expression levels of BpCBF9, BpCBF10 and BpCBF11 were significantly higher in roots than those in stems or leaves. All the related qRT-PCR data were listed in Appendix A.

### 2.7. Tissue-Specific Expression Pattern Analysis of BpCBFs under ABA Treatment

The *cis*-acting element analysis of the 20 BpCBFs proved the necessity of studying the responses of these transcription factors upon stress conditions. To investigate the responses of the 20 BpCBFs upon ABA treatment, 100 μM ABA treatment was performed for different times (0 h, 3 h, 6 h, 12 h, 24 h and 48 h) on birch seedlings and the relative expression levels of these genes in roots, stems and leaves were measured by qRT-PCR (Figure 8). The results showed that most BpCBFs were responsive to ABA treatment. The expression patterns of these genes under ABA treatment were different while that of each gene in different tissues is similar. The expression levels of most BpCBFs decreased since the ABA treatment began, reached the minimum at 12 h or 24 h time point and bounced back to a certain extent thereafter. And the minimum of some relative expression levels showed a delayed effect in stems and leaves compared to roots. Taking BpCBF1 as an example, its expression level in roots decreased since the ABA treatment, reached the minimum at 12 h, and then increased at 24 h and 48 h. BpCBF4 and BpCBF12 were significantly down-regulated in roots since the ABA treatment, expressed the lowest levels at 12 h and increasingly expressed at 24 h and 48 h. The expression levels of some BpCBFs did not change as obviously as those of other members, such as BpCBF7, BpCBF17 and BpCBF20. All the data of expression level (log_2_ value) were listed in Appendix A.

### 2.8. Tissue-Specific Expression Patterns Analysis of the BpCBFs under Salt Stress

To investigate the responses of the 20 BpCBFs upon salt treatment, 200 mM NaCl treatment was performed for different times (0 h, 3 h, 6 h, 12 h, 24 h and 48 h) on birch seedlings, and the relative expression levels of these genes in roots, stems and leaves were measured using qRT-PCR (Figure 9). The results showed that most of BpCBFs were up-regulated in different degrees after the salt treatment was performed, while the up-regulation reached the maximum at different time points and bounced back to a certain extent thereafter. The maximum of some expression levels showed an earlier effect in stems and leaves than in roots. Taking BpCBF1 as an example, its expression level in roots increased since the salt treatment began, reached the maximum at 12 h and then decreased at 24 h. BpCBF12 was the most expressed at 12 h time point in roots and then less expressed at 24 h. Some genes, such as BpCBF5 and BpCBF16, were not as responsive to salt treatment as other members. All the data of relative expression level (log_2_ value) were listed in Appendix A.

### 2.9. Identification of BpCBFs Responding to Cold Stress and Verification of RNA-Seq Data of Betula platyphylla under Cold Stress

For the verification of the RNA-seq data [38], the cold-responsive genes (BpCBF1, BpCBF2, BpCBF4, BpCBF7, BpCBF10 and BpCBF12) were selected for quantitative validation. As the results of tissue-specific expression pattern analysis showed, these genes were the most expressed in leaves, in which case their relative expression levels upon cold stress in leaves were determined by using qRT-PCR (Figure 10). These six genes were all significantly up-regulated upon cold treatment, and this trend is consistent with the result of the RNA-seq data analysis. The result of the RNA-seq data is displayed in Appendix A. The related qRT-PCR data are listed in Appendix A.

### 2.10. Transcription Activating Activity of BpCBFs Responsive to Cold Stress

The transcription activating activity of BpCBFs responsive to cold stress were investigated with BpCBF7 as a representative. A *BpCBF7* gene was introduced into a pGBKT7 vector, and the recombinant vector was transformed into yeast cell. The negative control did not grow on the nutrition-deprived culture medium added with AbA, while the yeast transformed with pGBKT7-BpCBF7 could normally grow on this kind of culture medium, like positive control did (Figure 11). This result showed that BpCBF7 had self-activation. To further narrow the range where the self-activation site was located, different lengths of *BpCBF7* sequence were introduced into the pGBKT7 vector. The result showed that the yeast transformed with pGBKT7-BpCBF7 (1–147) could normally grow on the nutrition-deprived culture medium added with AbA, while the yeast transformed with pGBKT7-BpCBF7 (1–146) could not. It can be deduced that the self-activation site was near the No. 147 amino acid site.

## 3. Discussion

Surviving in the changing environment, plants have to cope with all kinds of abiotic stresses with various molecular, biochemical and physiological methods, one of which is transcription regulation. CBF subfamily transcription factors, as important members of AP2/ERF transcription factor family, play important roles in plant stress response in many species [39]. In model plant *Arabidopsis*, six CBF transcription factors were identified and characterized in response to cold, drought or salt stress [4,5,6,11,12,13]. In woody plant *Populus trichocarpa*, six CBF transcription factors were identified [29], while their functions still remained undetermined. With the development of bioinformatic tools and sequencing, it is necessary to identify the CBF transcription factors in *Betula platyphylla*, a tree species with good adaptability and tolerance to abiotic stresses.

In this study, a total of 20 BpCBF transcription factor subfamily members were identified with AtCBFs as query sequences according to conserved domain characteristics. The increased subfamily members may explain the reason for the better defense of birch than *Arabidopsis* or poplar against abiotic stress. The basic properties and gene structures of BpCBFs were analyzed. As Figure 1D showed, *BpCBF14-20* shared the same chromosome location, more similar gene structure and more similar motif distribution like Motif 8, from which it can be inferred that these seven genes possibly are the result of gene duplication event and endowed to birch a stronger defense against abiotic stresses. The results of collinearity analysis showed that *BpCBF* had more syntenic transcription factor pairs with *PtCBF* than that of *AtCBF*. The *BpCBF14* had collinear genes in the other two plants, and there were collinear sites in the other two species, and only *BpCBF14* showed collinearity with the AtCBFs, which implied that BpCBF14 was more conservative and might play important roles in certain biological process. It is worth noting that CBFs in poplar have collinear genes in *Betula platyphylla* except PtCBF2, from which can be inferred that the evolution of CBF transcription factors is conservative and plays an indispensable role in woody plants. The *cis*-acting elements in the promoters of *BpCBF*s were identified, including environmental stress-related and hormone-related element components, such as abscisic acid response element (ABRE), stress response element (STRE), gibberellin response element (GARE) and low-temperature response element. It can be inferred that CBF transcription factors in birch can be closely related to abiotic stress response. In this case, the tissue-specific expression patterns of *BpCBF*s under ABA or salt treatment were analyzed using qRT-PCR. Most *BpCBF*s were down-regulated since the ABA treatment began, reached the minimum at 12 h or 24 h time point and bounced back to a certain extent thereafter. And the minimum showed a delayed effect in stems and leaves compared to roots. Taking *BpCBF1* as an example, its relative expression level in roots deceased since the ABA treatment, reached the minimum at 12 h and then increased at 24 h and 48 h. However, its relative expression levels in stems and leaves did not reach the minimum at 12 h. In contrast, most of *BpCBF*s were up-regulated since the salt treatment began, while the up-regulation reached the maximum at different time points and bounced back to a certain extent thereafter. The maximum showed an earlier effect in stems and leaves than in roots. Taking *BpCBF1* as an example, its expression level in roots increased since the salt treatment began, reached the maximum at 12 h and then decreased at 24 h. However, its expression levels in stems and leaves reached the maximum at 3 h time point. The delayed or earlier effect may be due to the different responses of different plant tissues against different abiotic stresses. Otherwise, the relative expression levels of some *BpCBF*s did not change significantly upon ABA or salt stress, which was possibly due to their involvement in responses to other stresses or other biological processes. The analysis of the transcription activating activity of the cold-responsive BpCBFs showed that BpCBF7 had transcription activating activity, and the transcription activating site was near the No. 147 amino acid. Due to the high sequence similarity of these six BpCBFs near this site, we deduced that this region may contribute to the transcription activating activity of all these six BpCBFs.

The results in this study proved that most of the 20 BpCBF transcription factors were responsive to ABA or salt treatment, in different expression patterns. The six cold-responsive BpCBFs were all up-regulated upon cold treatment, which was consistent with the result of RNA-seq. The analysis of the transcription activating activity of the cold-responsive BpCBFs can provide a reference for future research on BpCBF protein functions.

## 4. Materials and Methods

### 4.1. Identification and Analysis of CBF Transcription Factor Family in Betula platyphylla

The amino acid sequences of the 6 *Arabidopsis thaliana* CBFs (AT1G12610, AT1G63030, AT4G25470, AT4G25480, AT4G25490 and AT5G51990) were downloaded from the Phytozome v13.1 database (https://phytozome.jgi.doe.gov/pz/portal.html (accessed on 24 April 2023)). Two methods were used to screen and identify CBF subfamily members in *Betula platyphylla*. On the one hand, AtCBFs were used as query sequences to search for CBF sequences in *Betula platyphylla* through BlastP [40]. On the other hand, the HMM file of CBF (AP2: PF00847) was retrieved from the Pfam database (http://pfam.xfam.org/ (accessed on 26 April 2023)), and HMMER v3.1 [41] tool was used to obtain putative CBF family members from the *Betula platyphylla* database. Additionally, another two conservative domains were used for further verification according to Nie’s method [8]. Finally, we reserved the sequences of BpCBFs, including 20 members, that contained all the three conserved domains. Basic characteristics of the BpCBF amino acid sequences were analyzed using ExPASy (http://www.expasy.org/ (accessed on 27 April 2023)), such as molecular weight, isoelectric point, amino acid number, aliphatic index and hydrophilic mean (GRAVY) score.

### 4.2. Analysis of Gene Structure and Conserved Motifs

The gene sequences and CDS of BpCBFs were downloaded from Phytozome, and then analyzed with the online software GSDS 2.0 (Gene Structure Display Server: http://gsds.cbi.pku.edu.cn/ (accessed on 30 April 2023)) for exon and intron distribution patterns. The online software MEME 5.0 (http://meme-suite.org/ (accessed on 1 May 2023)) was used to predict the conserved motifs of the BpCBF protein sequences.

### 4.3. Multiple Sequence Alignment and Phylogenetic Analysis of the 20 BpCBFs

Multiple sequence alignment of amino acid sequences was performed using BioEdit (version 7.2.5), and the conserved motifs were checked using TBtools-II (v1.120) software. The phylogenetic tree (1000 bootstrap replications) was constructed with CBF subfamily members in *Betula platyphylla*, *Populus trichocarpa* and *Arabidopsis thaliana* via neighbor-joining (NJ) method using MEGA v7.0 software.

### 4.4. Chromosome Distribution and Gene Duplication

The location information of the BpCBFs was retrieved from Phytozome database and Tbtools-II (v1.120) was used to visualize the chromosome distribution. Gff3 (General Feature Format 3) files of *Betula platyphylla* v1.1, Arabidopsis thaiana TAIR10 and *Populus trichocarpa* v4.1 were searched and downloaded from Phytozome. Tbtools software was used to construct collinear analysis diagram and explore the gene duplication events of BpCBFs.

### 4.5. Analysis of Cis-Acting Elements

The 2000 bp sequence upstream of BpCBF CDS was collected from the Phytozome database (www.Phytozome.net, accessed on 6 May 2023) and was analyzed using the PlantCARE online service platform (http://bioinformatics.psb.ugent.be/webtools/plantcare/html/ (accessed on 6 May 2023)). The generated *cis*-acting elements were visualized by Tbtools.

### 4.6. Plant Materials, Growth Conditions, Treatments and Sampling

Birch (*Betula platyphylla*) seedlings were preserved by the State Key Laboratory of Tree Genetics and Breeding, Northeast Forestry University (Harbin). The seedlings for experiment use grew for 60 days in hydroponic culture (1/2 MS medium with 25 g/L sucrose, 0.02 mg/L NAA and 0.4 mg IBA) and then through homogenization. Seedlings were treated in hydroponic culture containing 100 μM ABA [42] or 200 mM NaCl [43]. Growth conditions were set with 16-h light/8 h darkness in a greenhouse at 25 °C. The ABA treatment time was 0 h, 3 h, 6 h, 12 h, 24 h, 48 h, and the salt treatment time was 0 h, 3 h, 6 h, 12 h, 24 h. The birch seedlings were grown in tissue culture containers for 60 days, then transplanted to soil, and grew for another 30 days for cold treatment at 4 °C for 3 h [38].

After ABA treatment and salt treatment, roots, stems and leaves of the plants were sampled, respectively. After cold treatment, only leaves of the plants were sampled. The nutrient solution on the surface was rinsed with deionized water, and then the excess water was absorbed with absorbent tissue. The collected plant tissue samples were immediately frozen in liquid nitrogen and stored in a refrigerator at −80 °C for subsequent analysis. In order to obtain reproducible results, there were three sets of biological replicates.

### 4.7. RNA Extraction and Expression Analysis

Total RNA was extracted from different birch tissues (roots, stems and leaves) using RNA extraction kit (Bioteke). RNA concentration was measured, and its quality was examined by electrophoresis with 1% agarose gel. Then, single-stranded cDNA was synthesized using reverse transcription kit (PrimeScriptTM RT reagent Kit, Takara Bio, Kusatsu, Japan). Quantitative primers were designed according to the downloaded full-length cDNA sequences of BpCBFs, and the internal reference gene was 18S ribosomal RNA [44].

The cDNA obtained by reverse transcription was diluted 10 times and used as the template for qRT-PCR. Based on SYBR Green fluorescence program, the qRT-PCR experiment was performed using THUNDERBIRD Next SYBR qPCR Mix (TOYOBO, Osaka, Japan). The total reaction volume was 20 μL. The specific reaction conditions were as follows: 95 °C for 5 min, then 95 °C for 15 s, 60 °C for 5 min for 45 cycles. The amplification reaction was carried out on Applied Biosystems 7500 Fast Real-Time PCR System. All reactions were repeated 3 times, and the analysis of relative expression levels of genes was carried out with the 2^−ΔΔCt^ method [45].

Primers used in qRT-PCR were designed by Primer 5.0 (Premier Biosoft, Palo Alto, CA, USA), listed in Appendix A. Due to high sequence similarity, a pair of primers was designed for the simultaneous quantification of BpCBF2 and BpCBF3, and another pair of primers for BpCBF9, BpCBF10 and BpCBF11. The heatmaps of tissue-specific expression analysis of BpCBFs were constructed by GraphPad Prism 9.0.0. Difference significances were analyzed with Duncan’s multiple range test method (*p* < 0.05).

### 4.8. Identification of Transcriptional Activation Activity of BpCBFs

To identify the transcriptional activation activity of the six cold-responsive BpCBFs, the BpCBF7 as a representative, was introduced into pGBKT7 yeast expression vector. The recombinant pGBKT7- BpCBF7, pGBKT-53/pGADT7-T (positive control) and pGBKT7 (negative control) were transformed into yeast competent cells (Y2H), which were then cultured on the nutrition-deprived yeast culture medium for 3–5 d at 30 °C. Added in the yeast culture medium, X-α-Gal (40 mg/L in culture medium) and AbA (500 ng/L in culture medium) were for the detection of whether yeast GAL4 system was activated.

## 5. Conclusions

In this study, a total of 20 CBF transcription factor subfamily members in *Betula platyphylla* were identified and can be classified into three clades. The conserved domains, physicochemical properties, phylogenetic relationship, exon-intron gene structure and the structure of conserved protein motifs of BpCBF were analyzed. The collinearity analysis of *CBF* genes was performed between *Betula platyphylla* and *Arabidopsis thaliana* and *Betula platyphylla* and *Populus trichocarpa*. The *cis*-acting elements in the promoter region of *BpCBF*s were identified, which were mainly related to environmental stress-related and hormone-related element components. The expression patterns of the 20 *BpCBF*s upon ABA or salt treatment showed that most of these transcription factors were responsive to ABA or salt stress, however, in different degrees in different tissues. The up-regulation of relative expression levels of the six cold-responsive genes was consistent with the result of RNA-seq. BpCBF7, as a representative of the six cold-responsive transcription factors, showed transcription activating activity. All these results indicate that CBF transcription factors in *Betula platyphylla* play a significant role in ABA, salt or cold response of birch.

## Figures and Tables

**Figure 1 ijms-24-10573-f001:**
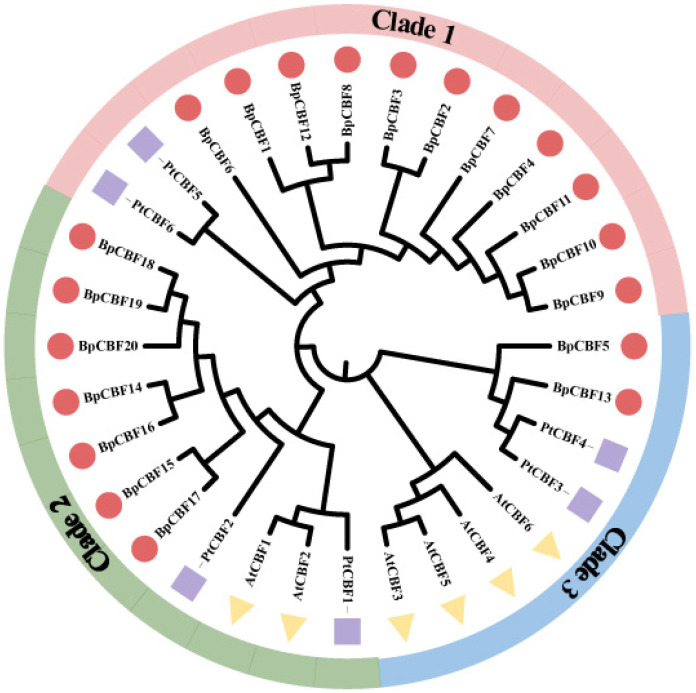
Phylogenetic analysis of CBF proteins from *Betula platyphylla*, *Arabidopsis thaliana* and *Populus trichocarpa*. The red circle represents BpCBFs; the yellow triangle represents AtCBFs; the purple square represents PtCBFs. Pink, green and blue represent Clades 1, 2 and 3, respectively.

**Figure 2 ijms-24-10573-f002:**
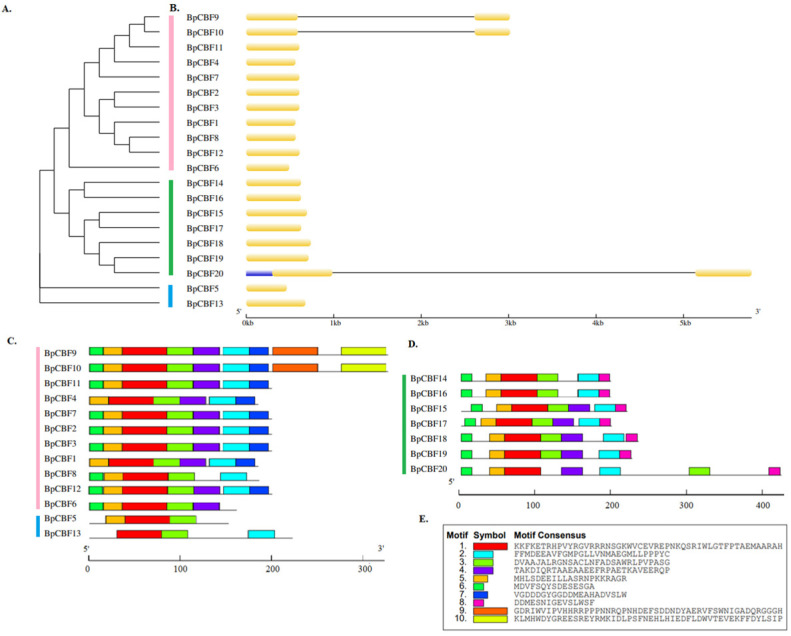
Phylogenetic relationship, exon-intron gene structure, structure of conserved protein motifs in BpCBF transcription factors. (**A**) Phylogenetic tree was constructed based on the full-length sequence of the BpCBFs protein. Pink line indicates Clade 1, green line indicates Clade 2 and blue line indicates Clade 3. (**B**) Exon-intron structure of the BpCBF transcription factors. Yellow boxes indicate exons, blue boxes indicate untranslated 5′- regions and black lines indicate introns. (**C**,**D**) are the motif composition of BpCBFs protein. The pattern of motif 1–10 is displayed with boxes in different colors. (**E**) Sequences of Motif 1–10. Motifs in the amino acid sequence of BpCBFs were predicted using online MEME tool.

**Figure 3 ijms-24-10573-f003:**
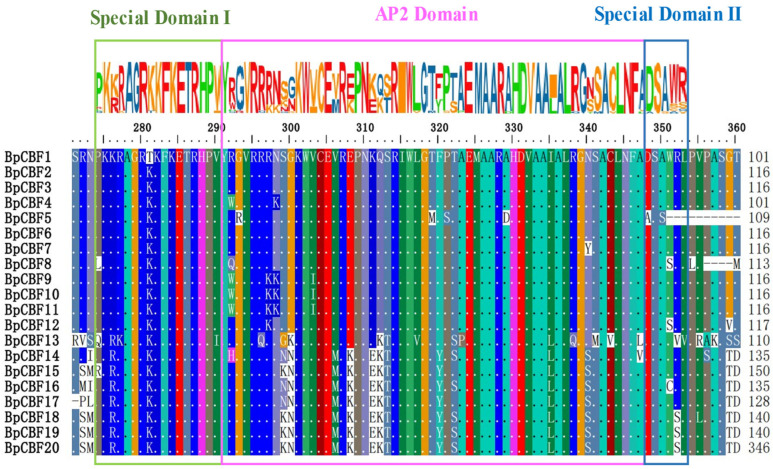
Partial multiple sequence alignment analysis of the 20 BpCBFs. The green box represents Special Domain Ⅰ; the pink box represents AP2 domain and the blue box represents Special Domain Ⅱ. The amplified colored letters above protein sequences indicate the motif of domains. Different colors of letters represent different amino acids.

**Figure 4 ijms-24-10573-f004:**
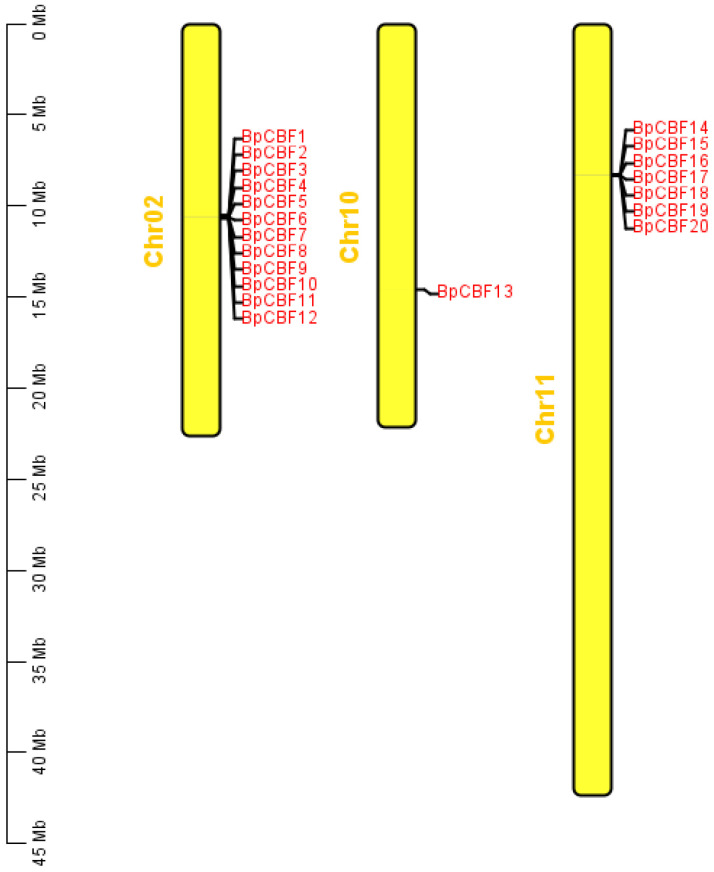
Chromosome location of the *BpCBF*s. The yellow square represents chromosome, and chromosome numbers are shown at the left side; Scale bar on the left represents the chromosome length (Mb).

**Figure 5 ijms-24-10573-f005:**
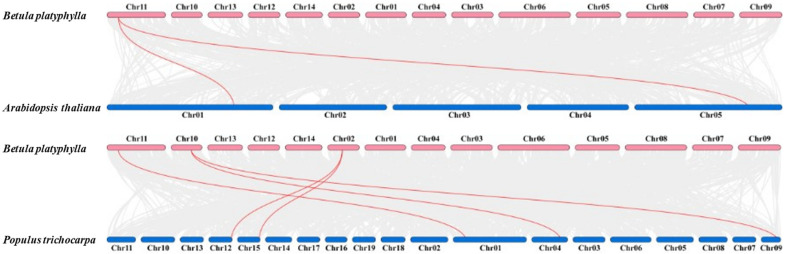
Collinearity analysis of CBFs between *Betula platyphylla* and *Arabidopsis thaliana* and *Betula platyphylla* and *Populus trichocarpa*. Grey lines in the background represent collinear blocks in *Betula platyphylla* and the other two plants, respectively. Pink lines represent collinear CBF transcription factor pairs. The numbers on the chromosomes indicate chromosome numbers.

**Figure 6 ijms-24-10573-f006:**
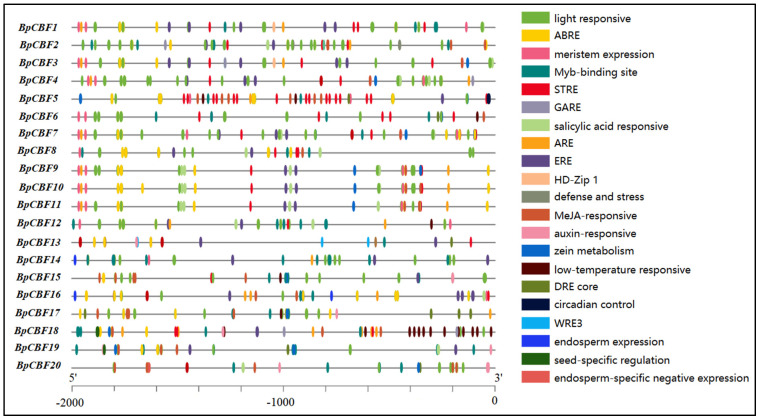
Analysis of *cis*-acting elements in the promoters of the 20 BpCBFs. Different colored boxes represent different *cis*-acting elements.

**Figure 7 ijms-24-10573-f007:**
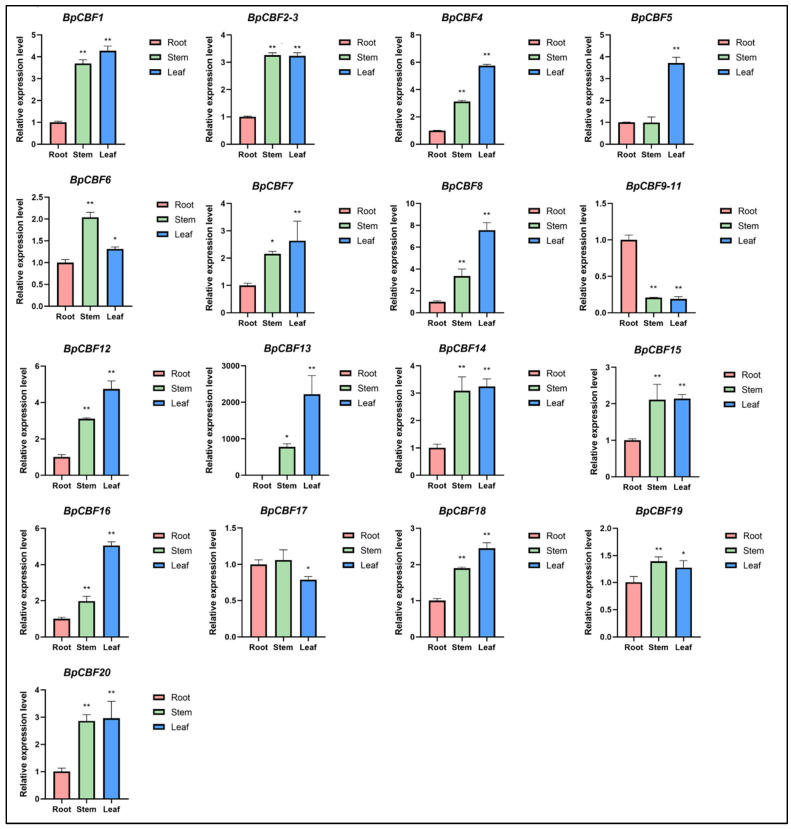
Tissue-specific expression analysis of BpCBFs. Tissue-specific expression analysis is performed with roots, stems and leaves of *Betula platyphylla*. The bar graphs were derived from the results of qRT-PCR. The expression level of each gene in plant roots was normalized to 1.0, and transcription levels were calculated using 2^−ΔΔCt^ method. (*t* test, * *p* < 0.05, ** *p* < 0.01).

**Figure 8 ijms-24-10573-f008:**
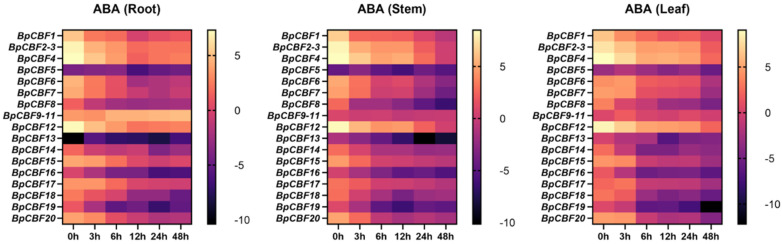
Tissue-specific expression analysis of *BpCBF*s under ABA treatment by qRT-PCR. The expression patterns of *BpCBF*s in roots, stems and leaves are shown. The concentration of ABA treatment is 100 μM, and the treatment time is 0 h, 3 h, 6 h, 12 h, 24 h and 48 h. The 2^−ΔΔCt^ method was used to calculate the transcription levels of *BpCBF*s, and the log_2_ (sample/control) value of each *BpCBF* was used to show its relative expression level. Scale bars were laid at the right side of heatmaps, and different colors indicated that the gene expression in the treated sample is up-regulated or down-regulated compared to the control.

**Figure 9 ijms-24-10573-f009:**
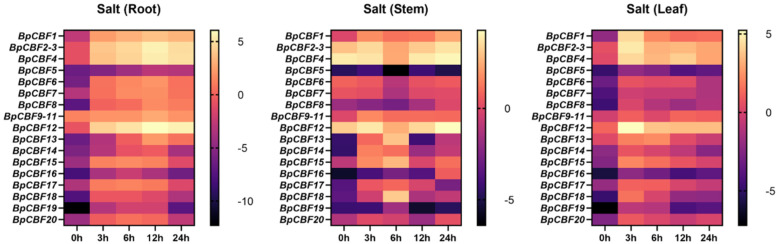
Tissue-specific expression analysis of *BpCBF*s under salt stress by qRT-PCR. The expression patterns of *BpCBF*s in roots, stems and leaves are shown. The concentration of salt treatment is 200 mM, and the treatment time is 0 h, 3 h, 6 h, 12 h, 24 h. The 2^−ΔΔCt^ method was used to calculate the transcription levels of *BpCBF*s, and the log_2_ (sample/control) value of each *BpCBF* was used to show its relative expression level. Scale bars were laid at the right side of heatmaps, and different colors indicated that the gene expression in the treated sample is up-regulated or down-regulated compared to the control.

**Figure 10 ijms-24-10573-f010:**
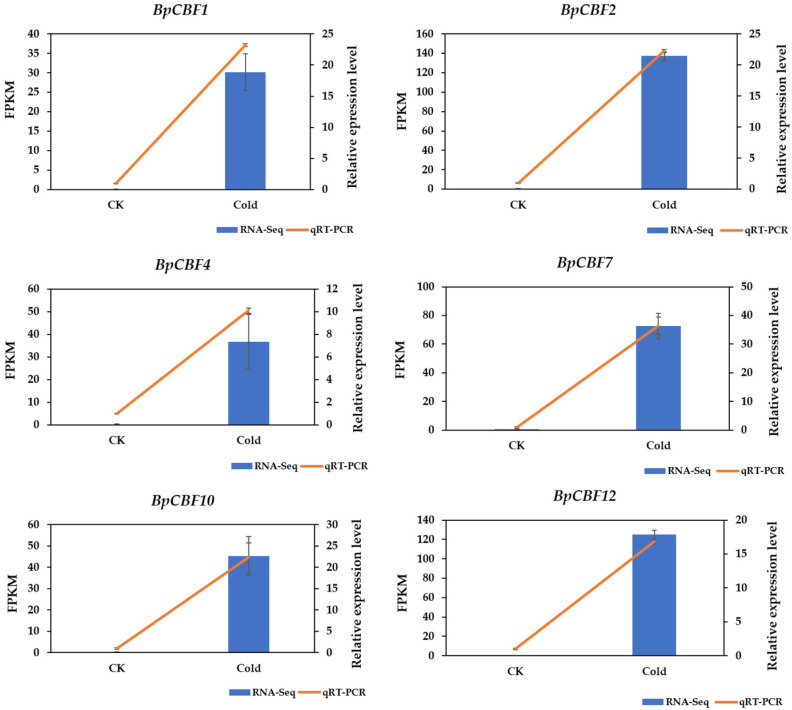
The expression analysis of BpCBFs responding to cold stress by qRT-PCR and RNA-seq data under cold stress in *Betula platyphylla* leaves. The temperature of cold treatment is 4 °C, and the treatment time is 3 h. The 2^−ΔΔCt^ method was used to calculate the transcription levels of six BpCBFs responding to cold stress, and the expression level of each gene in untreated plant leaves was normalized to 1.0. The blue column indicates the FPKM in the RNA-seq data, and the orange line indicates the qRT-PCR results. The left-side bar is the scale of FPKM in the RNA-seq data, and the right-side bar is the scale of expression level by qRT-PCR.

**Figure 11 ijms-24-10573-f011:**
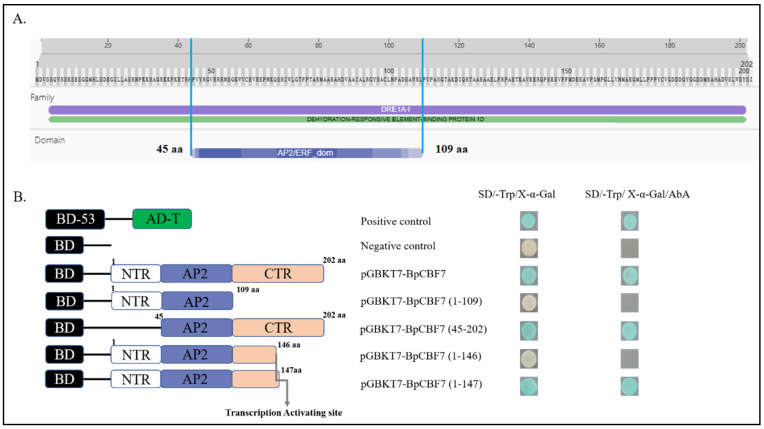
Identification of transcription activating activity of BpCBF7. (**A**) Display of the conserved domain of BpCBF7. The blue square under the protein sequence of BpCBF7 symbolizes the AP2 domain. (**B**) The localization of transcription activating activity site of BpCBF7 with yeast transformation method. BD means pGBKT7 vector. AD means pGADT7 vector. NTR means N-terminal of BpCBF7 and CTR means C-terminal. The number represent the site of amino acid (aa). The grey arrow shows the transcription activating site of BpCBF7. SD means nutrition-deprived yeast culture medium. The concentration of AbA in culture medium is 500 ng/L.

**Table 1 ijms-24-10573-t001:** Identification and physicochemical properties for the 20 *CBF* transcription factor family members information in *Betula platyphylla*.

Transcription Factor Name	Locus Name Phytozome v13	Genomic Serquence (bp)	Amino Acid No.	Molecular Weight (Da)	Isoelectric Points	GRAVY	Aliphatic Index	Chromosome Location
*BpCBF1*	BPChr02G23384	564	187	20,750.43	5.85	−0.459	68.93	Chr02:10503221..10503785 (+)
*BpCBF2*	BPChr02G23303	609	202	22,429.24	5.67	−0.515	64.36	Chr02:10522058..10522667 (+)
*BpCBF3*	BPChr02G23333	609	202	22,429.24	5.67	−0.515	64.36	Chr02:10531862..10532471 (+)
*BpCBF4*	BPChr02G23478	564	187	20,794.59	6.12	−0.488	66.36	Chr02:10541230..10541794 (+)
*BpCBF5*	BPChr02G23352	465	154	17,566.17	11.82	−0.719	61.49	Chr02:10582208..10582673 (+)
*BpCBF6*	BPChr02G23405	492	163	18,195.53	9.96	−0.702	56.99	Chr02:10584488..10584980 (+)
*BpCBF7*	BPChr02G23456	609	202	22,275.07	5.65	−0.478	65.30	Chr02:10595771..10596380 (+)
*BpCBF8*	BPChr02G23401	567	187	20,805.62	6.33	−0.438	65.53	Chr02:10606091..10606658 (+)
*BpCBF9*	BPChr02G23413	3015	331	37,548.72	5.39	−0.756	61.36	Chr02:10637802..10640817 (+)
*BpCBF10*	BPChr02G23391	3017	331	37,548.72	5.39	−0.756	61.36	Chr02:10644105..10647122 (+)
*BpCBF11*	BPChr02G23459	609	202	22,330.23	5.65	−0.440	66.29	Chr02:10650404..10651013 (+)
*BpCBF12*	BPChr02G23338	612	203	22,347.07	5.34	−0.499	64.53	Chr02:10660443..10661055 (+)
*BpCBF13*	BPChr10G03390	678	225	25,145.75	7.66	−0.345	82.31	Chr10:14620830..14621508 (−)
*BpCBF14*	BPChr11G06969	627	207	23,030.71	5.30	−0.507	66.25	Chr11:8279927..8280554 (−)
*BpCBF15*	BPChr11G06973	696	231	25,902.06	5.41	−0.560	60.04	Chr11:8285278..8285974 (−)
*BpCBF16*	BPChr11G07024	627	208	23,131.99	5.21	−0.504	65.77	Chr11:8298502..8299129 (−)
*BpCBF17*	BPChr11G06769	630	209	23,582.58	5.62	−0.559	64.02	Chr11:8313160..8313790 (−)
*BpCBF18*	BPChr11G06967	741	246	27,052.19	4.78	−0.455	66.34	Chr11:8319557..8320298 (−)
*BpCBF19*	BPChr11G06818	714	237	26,167.22	4.96	−0.467	67.22	Chr11:8329468..8330182 (−)
*BpCBF20*	BPChr11G06723	5779	444	49,211.02	5.12	−0.527	68.90	Chr11:8334005..8339784 (−)

Note: (+) means forward; (−) means reverse.

## Data Availability

Not applicable.

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
