# Peer review of "Genome-Wide Identification and Expression Profiles of C-Repeat Binding Factor Transcription Factors in Betula platyphylla under Abiotic Stress"

_ijms, 2023, doi:10.3390/ijms241310573_

Round 1

Reviewer 1 Report

The subject of the paper entitled " Genome-wide Identification and Expression Profiles of CBF Transcription Factors in Betula platyphylla under Abiotic Stress” by Zhang x., Yu J., Wang R., Liu W., Chen S., Wang Y., Yu Y., Qu G., and Chen S.,  is connected to genes in stress conditions. The plant species which serves as the research material, Betula platyphylla belongs to the species which occur naturally in subarctic and temperate Asia in the area of such countries like Japan, China, Korea, Mongolia, and Russia (Russian Far East and Siberia). This species, the Asian white birch or the Japanese white birch have become more popular in different regions (e.g. European parks) due to their ornamental value. Some of the reasons this tree has become more popular is its ornamental white trunk and main branches, wide, irregularly oval to linear crown, and very early (in comparison to the other birch species) sprouting.  Further, the tree has large, fresh green decorative leaves and relatively good resistance for wind.

According to the authors of the paper, research around the CBF (C-repeat binding factor) transcription factor subfamily which belongs to AP2/ERF (Apetala 2/ethylene-responsive factor) has not been studied up in the Asian birch. However, some CBF transcription factors have been identified in some plant species like the Arabidopsis thaliana, Nicotiana tabacum, Licopersicon esculentum and among tree species in Populus genus. Nonetheless, taking into consideration the vital role of the described transcription factor in the plant abiotic stress response, the research presented in the reviewed paper seems to be justified.

In the Introduction (section 1.) which is written in a typical fashion for scientific publication lists abiotic factors and briefly characterizes the CBF transcription factor subfamily. The authors emphasized the role this factor plays in the stress response. Most of this section is devoted to information about data connected with the study which were performed earlier on different plants, and these information are a good, short background to the presented results in the further part of publication.

The second part of the reviewed publication (i.e., 2. Results) gives the effects of the research obtained by authors. I would like to emphasize that this section is very large and the results are described in high levels of detail. The description of the results starts from the large table (table no. 1) entitled “Identification and physicochemical properties for the 20 CBF transcription factor family members information in Betula platyphyllos” and contains the key data connected the studied transcription factor by the authors. Further, the results are showed in the form of figures that document the large amount of results obtained by the authors. The Figures present the data in detail, they are also described in detail. It is worth emphasizing that the documentation in this section is of high quality. In addition to Table 1, the results are supplemented with 11 different Figures. The figures make the results very clear. I am convinced that the authors conducted the research described in the manuscript in a professional way. Undoubtedly, the results were preformed using a variety of methods according to the professional rules.

 In the relatively short, third section (i.e. 3. Discussion), the results presented in part 2. Results are thoroughly analyzed and are properly compared to literature data. I would like to underline that the literature items (total 42 items) were properly chosen and cited by the authors of the paper.

In the section, 4. Materials and Methods, the authors presented the methods which are shown in eight subsections. The methods are appropriate to the goals planned by the authors and to the research conducted in presented paper.

The very large results obtained by the authors are summarized in section five (5. Conclusion). The conclusion of the reviewed paper, despite the larger content than usual, present the achievements obtained during the conducted studies. They are properly formulated and adequate to scope of the research.

Taking into consideration the importance of genomic research and the role of the investigated transcriptional factor in the abiotic stress response in plants, I find this paper very valuable, interesting and likely to spur additional studies. The reviewed paper is suitable for publication in the “International Journal of Molecular Sciences”.

Author Response

Thank you very much for your fantastic summary of this manuscript. In your comment, more accurate description and richer examples were provided, from which we’ve benefited a lot and changed some expressions accordingly. Thank you again for your precious time and kind support.

Reviewer 2 Report

Thank you for sending me this interesting manuscript by Xiang Zhang, et al.  The authors identified a total of 20 CBF transcription factor subfamily members in Betula  platyphylla and then they could classify into two classes. Authors analyzed the expression patterns of the 20 BpCBFs under ABA or salt treatment showing that most of these transcription factors were responsive to ABA or salt stress at certain extent across different plant tissues. This study sheds light on the responses of BpCBFs to abiotic stress and provides a reference for further study on CBF transcription factors in woody plants.

There are issues that are listed in order, as follows:

1) Page 1, line 24 : “The up-regulation  of the six cold-responsive  genes by qRT-PCR…” rephrase the sentence.

2) Page 7, line 167 : used the correct scientific name format.

3) It would be interesting to analyze the expression level of certain transcription factors associated to other abiotic stress like salinity, desiccation, or pH. In case there is additional published information about this type of stress, please include it.

4) I would like to see the RNA extraction gel results and I noticed that author did not mention housekeeping genes as references to ensure changes on expression levels across different tissues are linked to the treatment affecting the transcription factor.

certain ideas were difficult to follow due to some english issues. English editing is required for a better understanding 

Round 2

Reviewer 2 Report

After the authors addressed the concerns, this manuscript would be accepted for publication in this journal